# Saponin-Rich Extracts and Their Acid Hydrolysates Differentially Target Colorectal Cancer Metabolism in the Frame of Precision Nutrition

**DOI:** 10.3390/cancers12113399

**Published:** 2020-11-17

**Authors:** Marta Gómez de Cedrón, Joaquín Navarro del Hierro, Marina Reguero, Sonia Wagner, Adrián Bouzas, Adriana Quijada-Freire, Guillermo Reglero, Diana Martín, Ana Ramírez de Molina

**Affiliations:** 1Precision Nutrition and Cancer Program, Molecular Oncology Group, IMDEA Food Institute, CEI UAM + CSIC, E-28049 Madrid, Spain; marina.reguero@imdea.org (M.R.); sonia.wagner@imdea.org (S.W.); adrian.bouzas@imdea.org (A.B.); Adriana.quijada@imdea.org (A.Q.-F.); 2Department of Production and Characterization of Novel Foods, Institute of Food Science Research (CIAL) (CSIC.UAM), 28049 Madrid, Spain; joaquin.navarrodel@uam.es (J.N.d.H.); guillermo.reglero@uam.es (G.R.); diana.martin@uam.es (D.M.); 3Sección Departamental de Ciencias de la Alimentación, Facultad de Ciencias, Universidad Autónoma de Madrid, 28049 Madrid, Spain; 4NATAC BIOTECH, Electronica 7, 28923 Madrid, Spain; 5Medicinal Gardens SL, Marques de Urquijo 47, 28008 Madrid, Spain; 6Forchronic, CANAAN Research & Investment Group, Agustín de Betancourt 21, 28003 Madrid, Spain

**Keywords:** fenugreek, quinoa, saponins, sapogenins, colorectal cancer, metabolic reprogramming, cell bioenergetics, precision nutrition

## Abstract

**Simple Summary:**

Colorectal cancer remains the second leading cause of cancer-related death worldwide, which is a situation that requires the continuous search for natural bioactive compounds that can mitigate its effects or can be used as co-adjuvants in the frame of precision nutrition. The elucidation of the mechanisms underlying the bioenergetics of colorectal tumorous cells as well as the modulation of lipid metabolism-related genes may greatly contribute to personalized therapies. We show that saponin- and sapogenin-rich extracts from fenugreek are able to strongly inhibit colorectal cancer cells’ growth, which is related to a reduction in the mitochondrial oxidative phosphorylation and the inhibition of aerobic glycolysis, in which each of the extracts target the molecular pathways related to lipid metabolism in a different manner. We propose that this outcome may have important implications on their potential use against colon cancer in the context of precision nutrition.

**Abstract:**

Saponins or their aglycone form, sapogenin, have recently gained interest as bioactive agents due to their biological activities, their antitumoral effects being among them. Metabolic reprogramming has been recognized as a hallmark of cancer and, together with the increased aerobic glycolysis and glutaminolysis, the altered lipid metabolism is considered crucial to support cancer initiation and progression. The purpose of this study was to assess and compare the inhibitory effects on colorectal cancer cell lines of saponin-rich extracts from fenugreek and quinoa (FE and QE, respectively) and their hydrolyzed extracts as sapogenin-rich extracts (HFE and HQE, respectively). By mean of the latest technology in the analysis of cell bioenergetics, we demonstrate that FE and HFE diminished mitochondrial oxidative phosphorylation and aerobic glycolysis; meanwhile, quinoa extracts did not show relevant activities. Distinct molecular mechanisms were identified for fenugreek: FE inhibited the expression of *TYMS1* and *TK1*, synergizing with the chemotherapeutic drug 5-fluorouracil (5-FU); meanwhile, HFE inhibited lipid metabolism targets, leading to diminished intracellular lipid content. The relevance of considering the coexisting compounds of the extracts or their hydrolysis transformation as innovative strategies to augment the therapeutic potential of the extracts, and the specific subgroup of patients where each extract would be more beneficial, are discussed in the frame of precision nutrition.

## 1. Introduction

Cancer is a complex and heterogeneous disease where genetic and environmental factors contribute to initiation, progression, and resistance to treatments. Metabolic reprogramming has been recognized as a hallmark of cancer [1,2]. Tumor cells frequently upregulate aerobic glycolysis (Warburg effect) and glutaminolysis to sustain cell proliferation. However, in recent years, alterations in lipid metabolism have been identified as key factors in cancer [2,3,4]. Lipids represent an important source of energy, as well as structural resources for the tumor, and they are also important signaling molecules for cellular pathways and intercellular communication. Although genetic alterations are the most studied factors, environmental factors including those related to lifestyle also contribute to the onset and progression of cancer. The World Health Organization (WHO) estimates that up to one-third of all cancers could be prevented through lifestyle changes, such as diet. In the case of gastric tumors, this percentage rises to more than half, constituting the main factor in their development [5]. Colorectal cancer (CRC) is the second most prevalent type of cancer in the world, and its appearance is closely associated to obesity and metabolic syndrome.

The recent development of powerful “omics” technologies (genomics, transcriptomics, proteomics, metabolomics, and lipidomics) has opened new avenues in nutritional sciences toward precision nutrition, which takes into account the differential responses to nutritional interventions based on gene variation (nutrigenetics), together with the scientific knowledge of the molecular targets and mechanism of action of bioactive compounds present in the diet (nutrigenomics). Moreover, precision nutrition also takes into account the nutritional and metabolic status of the individuals—obesity, T2DM (Type 2 Diabetes Mellitus), metabolic syndrome, mainly due to its association to chronic low-grade inflammation; life style parameters—alcohol consumption, diet, exercise; and the intestinal microbiome [6]. Precision nutrition can be considered as a therapeutic tool against chronic diseases related to metabolism, including cancer. This way, diet-derived ingredients, bioactive compounds from natural sources, and nutritional strategies can be developed as co-adjuvants in combination with the clinical therapies in cancer treatment [7,8]. 

Plants and food-derived ingredients constitute an inexhaustible source of bioactive compounds that may effectively inhibit the growth and progression of cancer cells and/or synergize with chemotherapeutic drugs used in the clinics, with fewer adverse and toxic effects. The success of the use of bioactive compounds in precision nutrition as co-adjuvants in the treatment of cancer patients requires the knowledge of their molecular targets and mechanism of action, which will identify the specific subgroup of patients where they should be implemented. Moreover, bioactive compounds may be effective not only against the signaling pathways involved in the tumor process but in alleviating other associated risk factors such as obesity, intestinal dysbiosis, and/or chronic inflammation [9,10,11,12,13].

Saponins constitute a wide group of structurally related compounds consisting of a non-polar aglycone—triterpenoid or steroidal sapogenins—attached to hydrophilic oligosaccharide moieties. Saponins are largely distributed in the plant kingdom and are mainly found in the seeds, leaves, roots, fruits, and stems. Triterpenoid saponins have been identified in legumes (soybean, lentils, alfalfa, and chickpeas, among many others), quinoa seeds, ginseng roots, quillaja bark, or liquorice roots, whereas steroid saponins have been found in fenugreek seeds, yucca, ginseng roots, asparagus, or oats [14]. Although saponins have traditionally been recognized as anti-nutrients, current research is being focused on saponins and sapogenins as bioactive compounds due their biological activities, including hypocholesterolemic effects, anti-inflammatory, immunomodulatory, antibacterial, antiviral and antitumor, among others [15]. 

The events that take place during the gastrointestinal digestion of saponins are highly related to their bioactivities. This is because, in general, it is assumed that saponins are poorly absorbed, showing a long residence time in the intestinal tract, and hence, their potential to cause biological activities at intestinal level is increased, as activity on CRC cells might be [16]. At the colonic level, the evidences have shown that saponins are transformed by the resident microbiota into sapogenins [17]. In this sense, it is important to remark that most sapogenins have demonstrated a superior bioavailability and higher concentrations at the systemic level, and it is generally assumed that some bioactivities of sapogenins are higher than those of their former saponins [16]. As an example, diosgenin (25*R*-spirosten-5-en-3β-ol) is the most studied steroidal sapogenin, fenugreek seeds being one of its richest sources, and excellent reviews have been done summarizing its biological activities [18]. Its antitumoral activities have been associated to the suppression of malignant transformation, reduction of the oxidative stress, regulation of T-cell antitumor immune response, inhibition of the epithelial–mesenchymal transition, remodeling of actin cytoskeleton, and inhibition of angiogenesis, among others [19]. As an example of a typical triterpenoid sapogenin, oleanolic acid is within the most popular ones, typically found in olive oil, but the saponins contained in quinoa seeds being recent sources of interest for this bioactive compound. The studies describing the antitumor activity of oleanolic acid have reported the inhibition of proliferation, induction of apoptosis, induction of autophagy, inhibition of infiltration, or cell cycle arrest [20,21]. 

Due to the potential of these molecules as bioactive agents, great efforts are being made to obtain saponin and sapogenin-rich extracts using non-conventional extraction methods, such as ultrasound-assisted extraction (UAE), which reduces time and energy consumption and augments the extraction efficiency. In a recent work, we produced UAE extracts from fenugreek and quinoa in order to obtain saponin-rich extracts from these popular edible seeds rich in both types of sapogenins, namely steroid and triterpenoid saponins (FE and QE), respectively [22]. Subsequently, the acid hydrolysis of these extracts was performed in order to obtain sapogenin-rich extracts (HFE and HQE), which was proposed as an innovative technological strategy to produce novel natural extracts with potential improved bioactivity. This is because, interestingly, in addition to sapogenins, different bioactive compounds of potential interest such as phytosterols, phenolic compounds, alkylresorcinols, tocopherol, or lipids were also identified in such extracts [22].

In the present work, we aim to compare the effects of saponin-rich extracts from fenugreek and quinoa (FE and QE) and their acid hydrolysates (HFE and HQE) on the cell viability of two colorectal cancer cells (DLD1 and SW620). The effect on cell bioenergetics was analyzed by measuring mitochondrial oxidative phosphorylation and aerobic glycolysis, as well as the expression of distinct lipid metabolism targets. We demonstrate that FE and HFE diminished mitochondrial oxidative phosphorylation and aerobic glycolysis, ending up with decreased ATP levels, whereas none of the quinoa extracts (QE and HQE) showed any relevant effect at the tested doses. Importantly, we have identified distinct molecular targets implicated in the mechanism of action of the fenugreek extracts. Finally, in the frame of precision nutrition, we discuss the relevance of coexisting compounds in the extracts as permeability enhancers, and the hydrolysis transformation of the extracts as innovative strategies to augment the clinical therapeutic potential of the extracts 

## 2. Results

### 2.1. Saponin- and Sapogenin-Rich Extracts from Fenugreek Inhibit Cell Viability of CRC Cell Lines

Saponins are a wide group of structurally related compounds formed by a non-polar aglycone—triterpenoid or steroidal sapogenins—attached to hydrophilic oligosaccharide moieties. For this reason, we first compared the effect of UAE extracts rich in saponins from two different sources—fenugreek (FE), rich in steroid saponins, and quinoa (QE), rich in triterpenoid saponins—on the inhibition of cell viability of CRC cell lines (DLD1 and SW620). Moreover, as sapogenins have been generally pointed out as superior bioactive compounds compared to their former saponins, and considering that the human gut microbiota has demonstrated to release such sapogenins [17], the acid hydrolysis of the saponin-rich extracts to obtain sapogenins-rich extracts HFE and HQE was performed to also assess their effect on cell viability. Dose–response curves and IC_50_ values after 48h of treatment with the different extracts are shown in Figure 1. For comparison purposes, commercial extracts from fenugreek (rich in steroid saponins) and quillaja (rich in triterpenoid saponins) were included.

QE showed no effects on the inhibition of the cell viability of CRC cell lines in the range of concentration assayed (0–120 µg/mL). HQE showed IC_50_ values close to 100 µg/mL. In line with these results, Qui, with saponins of the same triterpenoid nature, presented IC_50_ values in the similar range of HQE.

In contrast, FE, rich in steroid saponins, showed IC_50_ values below 20 µg/mL in both cell lines. HFE also showed significant effects on cell viability, although the IC_50_ values were 2.5 times higher than those of the non-hydrolyzed extract. 

Finally, the commercial fenugreek extract was less effective than the experimental FE, showing IC_50_ values >100 µg/mL. This extract contained 50% saponins, according to specifications. Unfortunately, neither a detailed and complete characterization of this extract in other compounds nor the procedure of extraction and fractionation of this commercial extract were available. 

### 2.2. Saponin- and Sapogenin-Rich Extracts from Fenugreek Diminish Cell Bioenergetics of CRC Cell Lines

In recent years, there is increasing interest on targeting the altered cancer metabolism. For this reason, and taking into account the interesting results observed in cell viability for the experimental fenugreek extracts, we aimed to investigate if the UAE saponin-rich extract FE and its hydrolysate HFE could target cancer cell metabolism, and more specifically the energetic metabolism (glycolysis and mitochondrial respiration) of colorectal cancer cells.

#### 2.2.1. FE and HFE Diminished the Mitochondrial Oxidative Phosphorylation of CRC Cell Lines 

First, we investigated the impact of saponin-rich extracts on the mitochondrial oxidative phosphorylation of colorectal cancer cells (DLD1 and SW620). By means of the Extracellular Flux Bioanalyzer (Seahorse Bioscience), we monitored the oxygen consumption rates (OCR) after the sequential addition of different drugs that regulate the mitochondrial function.

The study of cell bioenergetics (mitochondrial oxidative phosphorylation and aerobic glycolysis) was performed after treatment CRC cells for 48 h with different doses of FE and HFE extracts corresponding to 1/2 × IC_50_ (1/4 × LC_50_) and 1 × IC_50_ (1/2 × LC_50_). Non-treated cells were kept as controls. Importantly, before running the experiments, the same number of non-treated cells and pre-treated cells were plated in a XFe Seahorse plate in complete media (DMEM, 10% FBS) for 6 h to allow the cells to attach, without any treatment, in order to compare the cell bioenergetics only of viable cells (DLD1: 25,000 cells/well, SW620: 40,000 cells/well). Then, the medium of the cells was changed to the non-buffered XFe Base media supplemented with 10 mM glucose, 2 mM glutamine, and 1 mM pyruvate, and cells were incubated for 1 h at 37 °C without CO_2_. 

As it can be observed in Figure 2A (bioenergetic profile), DLD1 CRC cells pre-treated with FE and HFE showed reduced basal respiration rates (BRR) compared to control non-treated cells (measurements 1 to 3) at all the doses tested. After an injection of oligomycin, in order to estimate the OCR dedicated to ATP production, FE and HFE pre-treated cells displayed reduced levels of ATP compared to control non-treated cells (measurements 4 to 6). The maximal respiration rate (MRR) (measurements 7 to 9), after the injection of FCCP, was also affected in FE and HFE pre-treated cells. Finally, rotenone and antimycin A, inhibitors of complex I and III of the electron transport chain (ETC), respectively, were injected to shut down the OCR due to mitochondrial oxidative phosphorylation (measurements 10 to 12). These results indicate that FE and HFE clearly compromise mitochondrial respiration. Similar results were obtained in SW620 CRC cells (Figure 2B).

These effects were not observed in DLD1 CRC pre-treated cells with QE in accordance to the lack of effects on cell viability at the doses tested (Appendix A).

#### 2.2.2. FE and HFE Inhibit Aerobic Glycolysis of CRC

To analyze the effect of fenugreek extracts on aerobic glycolysis, we monitored the Extracellular Acidification Rate (ECAR), which is an indirect readout of the L-lactate production, after the sequential injection of modulators of the aerobic glycolysis. DLD1 or SW620 cells were pre-treated for 48 h with two different doses of FE or HFE corresponding to 1/2 × IC_50_ and 1 × IC_50_. Non-treated cells were kept as controls.

As it can be observed in Figure 3A (upper panel), basal ECARs of FE and HFE DLD1 pre-treated cells (1 to 3 measurements) were diminished compared to control non-treated cells. Next, we injected glucose (10 mM final concentration) to monitor cells’ ability to upregulate glycolysis when glucose is available. After the glucose injection, only HFE pre-treated cells had diminished levels of ECAR compared to control cells (3 to 6 measurements), indicating a reduced capacity for glycolysis. Then, oligomycin was injected to block the ATP production from mitochondria and so, to determine the Maximal Glycolytic Capacity. Pre-treated cells with FE and HFE at the higher doses (1 × IC_50_) diminished maximal ECAR levels compared to the control cells (measurements 6 to 9). 

Finally, 2-DG was injected (50 mM final concentration) to shut down aerobic glycolysis and to determine the non-glycolytic ECAR. 

Treatment SW620 cells with FE and HFE had little effect on inhibiting aerobic glycolysis (Figure 3B). Basal ECAR (measurements 1–3), quantified as the measurement 3, was not significantly diminished by the treatments. Nor the aerobic glycolysis (difference between basal ECAR and upregulated ECAR after glucose injection-measurements 4–6), nor the stressed ECAR (maximal ECAR after oligomycin injection-measurements 7–9).

FE and HFE had limited effect on targeting aerobic glycolysis of SW620 CRC cell lines (Figure 3B), reflecting different metabolic performance between DLD1 and SW620 CRC cell lines. 

As HIF1α is a key transcription factor promoting aerobic glycolysis, we also wanted to evaluate the effects of FE and HFE in the regulation of the expression of *HIF1A* and *LDHA*. No significant effects were found after treatment with FE or HFE, except for *LDHA*, which was significantly downregulated by FE at 30 µg/mL (*p* = 0.0264).

In summary, the analysis of cell bioenergetic profile of CRC cell lines pre-treated with FE and HFE extracts indicated that both extracts had a direct effect on diminishing bioenergetic pathways in CRC, with a higher impact on the inhibition of mitochondrial oxidative phosphorylation. 

Quantification of the intracellular ATP levels confirmed the effects of FE and HFE in the inhibition of CRC cell bioenergetics (Appendix A).

### 2.3. FE and HFE Inhibit CRC Metabolism by Targeting Distinct Molecular Pathways

The role of lipid metabolism alterations in cancer has been recently highlighted. In this regard, SREBP proteins, which are master regulators of lipid metabolism, together with FASN and SCD have been found to be upregulated in several neoplasias. Previous work by the group has identified the association between lipid metabolism genes and the prognosis of colon cancer patients, which provides a way to personalize the treatment of these patients [23,24]. 

In addition, current evidence shows that up to one-third of cancer deaths could be prevented by modifying risk factors, such as the consumption of saturated fatty acids. Moreover, the risk of death by cancer is increased 1.5–1.6 in individuals with a body mass index (BMI) > 40 kg/m^2^ [25]. Obesity induces a low grade of chronic inflammation promoting the carcinogenic process, as well as diminishing the response to antitumor treatments [26,27]. 

With the objective of evaluating the effect of FE and HFE on CRC cancer metabolism and risk factors related to obesity and inflammation, we designed a panel of metabolic genes—including de novo lipogenesis and cholesterogenesis (*SREBF1, FASN; SCD; SREBF2, HMGCR*), fatty acid metabolism (*ACSL1, ACSL4, SCD*), oncogenic pathways in CRC (*CHOKA, BMP2*), exogenous uptake of lipids (*LDLR, FABP1*), cholesterol metabolism (*ABCA1, ApoA1*), inflammation and oxidative stress (*JAK1, NEF2L2*), and resistance to chemotherapy (*TK1, TYMS*). In order to validate the specificity of the effect of the different extracts, we included in the analysis a supercritical rosemary extract (RE), whose molecular effects on these metabolic targets have been previously analyzed [7]. 

Interestingly, FE and HFE differently modulated the expression of lipid metabolism genes in DLD1 CRC cells pre-treated for 48 h with the extracts (Figure 4). 

FE diminished the expression of *ABCA1* implicated in the reverse transporter of cholesterol, which has been associated with poor prognosis in CRC patients [4]. Interestingly, FE augmented the expression of genes associated to the exogenous uptake of fatty acids and low-density lipoproteins (LDL), as well as genes related to de novo lipogenesis and cholesterogenesis. These results suggest that FE may promote anabolism, which may be of interest in cancer patients developing cachexia. In addition, FE significantly downregulated genes related to the synthesis of nucleic acids—*TYMS* and *TK1*, which have been implicated in the appearance of resistance to the anti-metabolite 5-fluorouracil [28]. In order to determine if FE may synergize with the chemotherapeutic drug 5-fluorouracil (5-FU), we pre-treated CRC cells for 4 h with FE (5 µg/mL corresponding to 1/4 of the IC_50_) before the addition of 5-FU at different doses: 5, 10, and 15 µg/mL. The combination index indicated a positive synergism between FE and 5-FU (Appendix A). 

On the other hand, HFE diminished lipid metabolism targets implicated in de novo lipogenesis and cholesterogenesis (including *SREBF1, FASN, SCD, SREBF2,* and *HMGCR*), and two metabolic axes—*ABCA1* [4] and *ACSL1/ACSL4/SCD* network [3]—were described to promote invasion and migration and to correlate with poorer prognosis in CRC patients. Moreover, *FABP1*—free fatty acid binding protein—and *BMP2* were also downregulated.

In line with these results, DLD1 CRC cells treated with HFE for 48 h were less prone to accumulate neutral lipid droplets compared to control non-treated cells (DMSO) (Appendix A). 

Both extracts augmented the expression of *GCNT3*, which has been described as a good prognosis biomarker in CRC [29], and *ApoA1*, described to counteract the protumoral effects of ABCA1 overexpression [4]. 

These results were validated in a second CRC cell line SW620 (Figure 5).

## 3. Discussion

The increase in the incidence of chronic diseases associated to metabolism such as obesity, insulin resistance, cardiovascular diseases, and cancer is a social and medical matter of concern. Related to cancer, evidence from epidemiological and clinical trials have shown that up to one-third of cancer deaths could be prevented by modifying key risk factors, diet and exercise being among the most important factors, due to their association with obesity [30,31]. The altered systemic metabolic function during obesity, including low-grade chronic inflammation, insulin resistance, hypercholesterolemia, and hypertriglyceridemia, seem to be on the basis of the association between obesity and the carcinogenic process. 

The reprogramming of metabolism is a key event in tumorigenesis [1], and efforts are being oriented to identify molecular targets and oncogenic pathways sustaining the altered metabolism in cancer. Together with the promotion of aerobic glycolysis (Warburg effect) and the increased glutaminolysis, the relevance of lipid metabolism in cancer has been extensively demonstrated [3,32,33]. Lipids, in addition to their role as fuels for ATP production, play a key role as components of plasmatic and organelles membranes, affecting their plasticity, and they are also important as cell signaling molecules. Lipid metabolism alterations have been associated with tumor progression and malignancy. In this regard, SREBP proteins, which are master regulators of lipid metabolism, together with FASN and SCD have been found to be upregulated in several neoplasias [32]. 

According to GLOBOCAN 2018 data, CRC is the third most deadly and fourth most commonly diagnosed cancer in the world [34]. In the course of this multifactorial condition, numerous alterations occur both in tumor suppressor and oncogenic genes. Experimental and epidemiological studies support the role of lipid metabolism, diet, gene–diet interactions, obesity, and gut microbiota in the etiology and prognosis in CRC [35].

Key enzymes of lipid metabolism are differentially expressed in normal and tumor CRC tissues. Some of them are associated with cancer survival and have been individually proposed as prognosis markers [35]. Lipid metabolism alterations not only affect the primary tumor but also influence malignancy, migration, invasion, relapse, and resistance to chemotherapy treatment. A lipid-metabolic signature ColoLipidGene is associated with CRC prognosis in stage II patients [23]. A lipid metabolic network formed by ACSL1, ACSL4 (Acyl-CoA synthetases), and SCD promotes migration and invasion by CRC cells [3]. Epidemiological studies have linked cholesterol levels with the risk of developing different tumors [36,37,38]. The formation of high-density lipoproteins (HDL) for reverse cholesterol transport is based on the action of the ABCA1 transporter, which has been recently associated with poor prognosis in colon cancer patients [4]. In addition, genetic analysis of 57 SNPs located in seven lipid metabolism locations has shown that rs8086 SNP in ACSL1 was associated with reduced mRNA expression levels of ACSL1 and shorter CRC disease-free survival (DFS) [39]. In addition, evidence has demonstrated that nearly half of the cancer deaths could be prevented by modifying key risk factors related to lifestyle (smoking, alcohol consumption, high saturated fatty acid intake) [40,41].

Efforts are being oriented to develop therapeutic strategies against cancer metabolism. Vegetables and plants are among the most popular sources to obtain bioactive compounds. Phytochemicals exert numerous biological activities, such as anti-inflammatory, antihypertensive, antioxidant, anticarcinogenic, antidiabetic, or antiobesity [42]. Thus, targeted nutritional interventions based on the knowledge of the molecular mechanism of action and molecular targets of bioactive compounds and/or ingredients from diet may contribute to enhance cancer therapy and the survival of cancer patients. 

In this sense, we recently produced extracts from the edible seed fenugreek (FE, HFE) and quinoa (QE, HQE) with a relevant content of different bioactive compounds, including saponins and sapogenins within the major bioactive components of the non-hydrolyzed and hydrolyzed extracts, respectively. Additionally, these extracts also contained other interesting compounds whose contribution to the final bioactivity might be of interest [22]. In general, saponin-rich extracts from both seeds also contained amino acids, glycerides, and carbohydrates, as well as phytosterols, tocols, and phenolic compounds. On the contrary, both sapogenin-rich extracts mainly contained free fatty acids and partial glycerides, and they were richer in phytosterols, phenolics, and tocols, and they did not contain any amino acids or carbohydrates. All four extracts have also shown promising effects on lipid absorption by inhibiting the pancreatic lipase, whereas both hydrolyzed extracts have very recently demonstrated to interfere with the bioaccessibility of cholesterol [43]. Even FE has shown a potential prebiotic-like effect [17]. Hence, the demonstrated multibioactivity of these extracts at different levels of the gastrointestinal tract motivated the further evaluation of these extracts as potential anticancer agents in colorectal cancer as an additional potential bioactivity at the gastrointestinal level. The suggested long residence time of saponins at this tract as well as the demonstrated biotransformation of saponins to sapogenins by microbiota were motivators as well [17]. 

As shown in Figure 1, QE showed no effects on the inhibition of the cell viability of CRC cell lines in the range of concentration assayed (0-120 µg/mL). HQE showed IC_50_ values close to 100 µg/mL. In line with these results, commercial quillaja (Qui), containing saponins of the same nature as those in quinoa, presented IC_50_ values in the similar range of HQE extract. These results suggest that QE and HQE extracts have little impact on the inhibition of proliferation of CRC cells (DLD1 and SW620). 

In contrast, FE, rich in steroid saponins, showed IC_50_ values below 20 µg/mL. HFE also showed significant effects on cell viability, although the IC_50_ values were 2.5 times higher than those of the non-hydrolyzed extract. 

Despite the fact that the HFE extract was the one that contained free sapogenins, which are considered as the most bioactive derivatives, the higher effect observed with FE suggests the presence of additional compounds not present in the HFE, acting synergistically with the steroid saponins on the inhibition of CRC cell viability. Additionally, the concentration of active compounds in both extracts should be also considered. The previous detailed characterization of FE and HFE showed important differences both in the phytochemical composition and in the concentration of saponins and sapogenins [22]. Thus, whereas saponins were one of the major components quantified in FE (31%), the sapogenin content of HFE was much lower (8%). FE also contained amino acids and carbohydrates, which were not present in case of HFE because, in general, HFE was mainly a less polar extract that was especially rich in free fatty acids (35%) such as linoleic and oleic acids and partial glycerides [22]. In this sense, it is interesting to remark the recent study performed by Storniolo et al. [44] where oleic acid promoted mitogenic activity in colon cancer cell lines, whereas olive oil, rich in this fatty acid, demonstrated protective effects against this disease. These contradictory results have been attributed to the protective effects of additional minor bioactive components present in olive oil, which seem to counteract the mitogenic effects of oleic acid alone. Similarly, it can be hypothesized that the minor bioactive components of HFE, such as sapogenins alone and/or in synergism with co-existing compounds, might be actively exerting antiproliferative effects regardless of the high free fatty acid content. In agreement, the exceedingly lower content of linoleic and oleic acids in FE (1.44%) might help partially explaining the superior bioactivity of this extract respect to HFE. Further studies in this respect might be worth exploring in order to elucidate the effects of co-extracted components on the antiproliferative activity of these extracts. 

By means of the latest technology in the analysis of cell bioenergetics, we demonstrated that FE and HFE target CRC cell bioenergetics by diminishing mitochondrial oxidative phosphorylation (Figure 2A,B) and aerobic glycolysis (Figure 3A,B), ending up with a decrease in the levels of ATP (Appendix A). As far as our knowledge goes, no previous studies have been performed in which the cell bioenergetics have been assessed with extracts where saponins and sapogenins are within the major bioactive components or with fenugreek extracts in general. Only a couple studies have very recently the steroid saponin gracillin as a mitochondria-targeting antitumor drug, thanks to its ability to induce a dose-dependent decrease of mitochondrial respiration and the inhibition of aerobic glycolysis [45,46]. 

Importantly, we have identified in this study distinct molecular targets implicated in the mechanism of action of the extracts. FE inhibited the expression of *TYMS* and *TK1*, which have been associated to the appearance to resistance to 5-FU. FE also synergized with 5-FU on the inhibition of CRC cell viability (Appendix A). Interestingly, we also found that FE upregulated lipid metabolism targets, such as *SREBF1, FASN, SCD SREBF2, HMGCR,* and *LDLR,* which suggest an anabolic effect for this extract. Very recently, Mohammad-Sadeghipour, Mahmoodi, Karimabad, and Mirzaei [47] found that a hydroalcoholic extract obtained from fenugreek also upregulated the expression of *LDLR* in SW480 colorectal cancer cell lines, although the expression of *FASN* was downregulated by the extract, as opposed to our findings. However, Zhout et al. [48] did find similar outcomes in healthy tissues. Specifically, an alfalfa saponin extract was able to upregulate the expression of *FASN* in hepatic tissue from laying hens. The upregulation of *LDLR* by an alfalfa saponin extract has also been demonstrated in healthy buffalo rat liver cells [49]. These results highlight the relevance of additional bioactive compounds of saponin-rich extracts contributing to define the specific targeted lipid metabolism network. 

In addition, FE upregulated the expression of *GCNT3*, which has been proposed as a biomarker of good prognosis [29,50], and *ApoA1*, which has shown to counteract the effects of ABCA1 in the promotion of invasiveness and stemness [4]. No previous studies have been found regarding the upregulation of *GCNT3* by fenugreek or saponins, nor the upregulation of *APOA1*.

On the other hand, HFE inhibited lipid metabolism targets, including *SREBF1, FASN, SCD, SREBF2, HMGCR, LDLR*, *FABP1*, *ABCA1,* and the metabolic axis *ACSL/SCD,* all of which have been implicated in poorer prognosis in CRC [3]. In line with these results, CRC cells treated with HFE showed a reduced lipid content compared to control non-treated cells (Appendix A). Mohammad-Sadeghipour, Mahmoodi, Karimabad, and Mirzaei [47] have also described the downregulation of the expression of *FASN FAS* by diosgenin, which appears to be the major aglycone in the HFE, although the expression of *LDLR* was upregulated by this sapogenin, as opposed to our findings. These results indicate that additional bioactive compounds are contributing to define the specific lipid targets modulated by whole extracts. 

It is important to remark that bioactive compounds exert their cytotoxic activity at specific regions of the gastrointestinal tract, depending on the bioaccessibility and cellular uptake at the intestinal level. In this sense, low bioaccessibility and cellular permeabilities would be required at the small intestine level if the targeted cancerous tissues are located in the last parts of the intestinal tract, such as colon. In general, a low intestinal permeability and poor bioavailability has been described for saponins, suggesting that these compounds would mainly reach the colon intact to exert their cytotoxic effect locally. In the case of sapogenins, they are also considered to have low bioavailability due to low aqueous solubility and poor bioaccessibility. However, these general ideas should not be widely assumed, since contrary outcomes have been observed in this regard. Thus, in the specific case of the experimental fenugreek extracts assayed, we previously demonstrated that saponins from FE had a complete bioaccessibility [22] and high in vitro membrane permeability [51] after the simulation of gastrointestinal digestion of this extract, which suggested that saponins from these extracts would be likely absorbed at first sections of the intestinal tract. This result was explained by the coexistence of other compounds in the extracts that might be acting as “permeability enhancers”. Specifically, coexisting lipids in the extracts were demonstrated as one of the permeability enhancers of saponins of FE [51], which provide new possibilities for the delivery of these bioactive compounds at different parts of the intestinal tract: (i) if it is required for saponins to act at the systemic level, the presence of co-existing permeability enhancers should be sought to augment permeability at the small intestine, as it was the case of FE; (ii) if it is required for saponins to target cancerous tissues located in the last parts of the intestinal tract, such as the colon, a low intestinal permeability and poor bioavailability should be sought. Considering these approaches, our results suggest that in the case of CRC patients developing cachexia, the high bioaccessibility and permeability of saponins from FE at the intestinal level (small intestine) would allow this extract to be a good candidate to promote anabolic effects systemically, taking into account the relevant results observed for *SREBF1, FASN, SCD,* and *LDLR*, among others, for this extract. On the other hand, in the case of CRC patients where lipid metabolism-related genes associated to progression and metastasis such as *SREBF1, FASN, ABCA1,* and the *ACSL/SCD* axis, among others, are overexpressed, it should be desirable to develop formulas to deliver FE to the colon in order to be metabolized by the colonic microbiota to release the aglycone sapogenins. In this sense, recent studies have confirmed that the human gut microbiota is capable of releasing the aglycones from saponins contained in FE [17], which is a situation that might allow sapogenins to act locally on colon cancer cells. However, alternatively to the development of colon-delivery formula, the production of HFE would be also an interesting strategy to act on such lipid metabolism genes at the colon level. This is because our previous studies showed that despite sapogenins from HFE having a good bioaccessibility [22], these compounds showed a poor in vitro membrane permeability [51]. Therefore, it might be expected that sapogenins from HFE would reach the last sections of the intestinal tract, allowing to exert their local effect on lipid metabolism-related genes in CRC cells. Nevertheless, further studies would be necessary in order to evaluate whether additional strategies are necessary to favor their permeability at the specific site of colon cells. According to our previous study, co-existing lipids would also be proper permeability enhancers for fenugreek sapogenins. 

Therefore, the distinct metabolic mechanisms of FE and HFE in targeting CRC metabolism suggest distinct scenarios in combination with specific technological strategies, such as the hydrolysis of extracts or formulations, in order to deliver the bioactive compounds at specific regions of the gastrointestinal tract to exert their bioactivity.

Finally, in the frame of precision nutrition, additional aspects have to be taken into consideration, as they will refine the final responses to specific nutritional intervention, such as the composition of the intestinal microbiome, the nutritional status of individuals, and/or the metabolic alterations at the systemic levels (obesity, T2DM, CDV, cancer…). Bioactive compounds and ingredients from the diet have a direct effect on the composition and diversity of the intestinal microbiota, as demonstrated for the assayed FE [51]. Importantly, numerous metabolites derived from the microbiota have been shown to regulate local inflammation as well as the innate and adaptive immune responses. In these sense, additional challenges are oriented to understand how diet might impact cancer development through modulating the composition of gut microbiota, the metabolites derived from microbiota, and the intestinal immune system functionality. Therefore, further studies will be oriented to explore whether the sapogenins released from FE by the microbiota, which has been demonstrated previously, and the modulation of the composition of microbiota by FE [51] would impact local inflammation and immune responses. The equilibrium between inflammation and tolerance in the gut has been shown to modulate precancerous lesions to progress or not into cancer [52]. In a similar way, the nutritional status of individuals (obesity, dyslipidemias) also impacts on the differential responses of individuals by triggering low-grade chronic inflammation and affecting the systemic energy balance. Numerous studies describe the role of chronic inflammation and the immune system in the carcinogenic process, as well as in the response to antitumor treatments. In this sense, the possibility of modulating the innate immune response, which stems at the intestinal level, through nutrition is receiving great interest in the treatment of cancer. Proper manipulation of this type of response could enhance other antitumor therapies and immunotherapies [53].

## 4. Materials and Methods 

### 4.1. Saponin-Rich Extracts and Sapogenin-Rich Extracts

Seeds of red quinoa (*Chenopodium quinoa*) were purchased from Hijo de Macario Marcos (Salamanca, Spain) and seeds of fenugreek (*Trigonella foenum-graecum*) were purchased from Murciana de Herboristeria (Murcia, Spain). 

The preparation of saponin-rich extracts was previously performed by Navarro, Reglero & Martin [22] by UAE with methanol and subsequent concentration of saponins with water and 1-butanol. The final dried extracts showed a saponin content of 31% for FE and 7.9% for QE, according to the HPLC-DAD analysis [22]. Subsequently, sapogenin-rich extracts were also obtained in the same previous study by acid hydrolysis and followed by an extraction of sapogenins with ethyl acetate. The final dried extracts showed a sapogenin content of 8% for HFE and 5.4% for HQE, according to the GC-MS analysis [22]. 

For comparative purposes, quillaja extract (20% *w/w*) (Qui) from Sigma Aldrich and commercial fenugreek extract from Biotanica (50% *w/w*, Aucklan, New Zealand) were used as reference commercial products of triterpenoid and steroid saponin-rich extracts, respectively. Supercritical-rosemary extract (RE) (rosemary antioxidant extract from Flavex 027.020) was also used for comparative purposes.

### 4.2. Cell culture

Colorectal cancer cell lines, DLD1, and SW620, were obtained from the American Type Culture Collection (Manassas, VA, USA) and were cultured in DMEM media supplemented with 10% fetal bovine serum (LONZA Iberica, S.A, Barcelona, Spain) in an incubator with 95% humidity and 5% CO_2_.

### 4.3. Cell Viability

The cytotoxic and antiproliferative activities of the different extracts in human CRC cell lines were determined by the MTT assay. Briefly, cells in the exponential growth phase were plated in 96-multiwell plates. After 24 h, media was replaced with 200 μL media containing serial concentrations of each extract (dissolved in DMSO) for 48 h. The number of viable cells was determined at time zero (control growth wells) and after treatments. To determine the number of viable cells, tetrazolium MTT salt solution (Sigma) (5 mg/mL in phosphate-buffered saline) was added for 3 h. Then, the formazan produced in each well was solubilized by adding 200 μL DMSO and measured using a spectrophotometer reader (λ  =  560 nm) (Biochrom Asys UVM 340 Microplate Reader; ISOGEN). Parameters for 50% of cell viability inhibition (IC_50_) were calculated accordingly to NIH definitions using a logistic regression and are indicated in Figure 1 [54]. 

The synergism between 5-FU and FE or HFE extracts was analyzed by the combination index (CI) obtained using the Calcusyn software (version 2.1, Biosoft, Cambridge, UK), based on the Chou–Talalay method [55].

### 4.4. Extracellular flux Analysis of the Oxygen Consumption Rate (OCR) and Extracellular Acidification Rate (ECAR): MitoStress and GlycoStress Tests

Mitochondrial oxidative phosphorylation (Cell MitoStress Test) and aerobic glycolysis (Cell GlycoStress Test) were monitored with the XFe96 Cell Bionalyzer (Seahorse Biosciences, XFe96). Optimal cell density and drugs tritation were previously determined.

The dependency of the cells for aerobic glycolysis and oxidative phosphorylation were monitored after the injection of several modulators of both bioenergetic pathways.

Prior to the experiments, cells were pre-treated with different doses of FE and HFE (1/2 × IC_50_ and 1 × IC_50_) for 48 h. Non-treated cells were kept as controls. 

For MitoStress assay, 25,000 DLD1 or 40,000 SW620 cells were plated in a XFe-96 well plate and cells were kept for 6 h in DMEM 10% FBS to allow the cells to attach. Then, medium was changed to 10 mM glucose, 2 mM glutamine, and 1 mM pyruvate XFe DMEM (5mM HEPES), and cells were incubated for 1 h at 37 °C without CO_2_. Three different modulators of the mitochondrial respiration were sequentially injected. After basal OCR determination (1 to 3 measurements), oligomycin (1.5 μM), which inhibits ATPase, was injected to determine the amount of oxygen dedicated to ATP production by mitochondria, (3 to 6 measurements). To determine the maximal respiration rate (MRR) or Spare Respiratory Capacity, FCCP (carbonyl cyanide-4-(trifluoromethoxy) phenylhydrazone) was injected (0.8 μM for DLD1 or 0.5 μM for SW620) to free the gradient of H^+^ from the mitochondrial intermembrane space (7 to 9 measurements), and thus, to activate maximal respiration. Finally, antimycin A and rotenone (0.5 μM) were added to completely inhibit mitochondrial respiration (10 to 12 measurements).

GlycoStress assay: First, 20,000 DLD1 cells or 30,000 SW620 cells were plated in a XFe 96-well plate and kept for 6 h in DMEM 10% FBS to allow the cells to attach. Then, culture medium was changed to 0.2 mM glutamine XFe DMEM (5mM HEPES) to starve the cells for 1h. First, basal ECAR was measured (1 to 3 measurements). Then, 10 mM glucose was injected (4 to 6 measurements) to the cells to determine glycolysis (this is the increased ECAR from the basal starved situation after glucose addition) and calculate the cells’ capacity to use glucose. Next, maximal glycolytic capacity was monitored (7 to 9 measurements) after an injection of oligomycin, which inhibits the ATP production from the oxidative mitochondrial respiration. Finally, a third injection with 50 mM DG was done to specifically shut down aerobic glycolysis (10 to 12 measurements).

### 4.5. Measurement of Intracellular ATP Content

For the quantification of the ATP content, the ATP-based assay CellTiter-Glo Luminescent Cell Viability kit was used (Promega, Madison, WI, USA; Cat # G7571) following manufacturer´s recommendations. Briefly, DLD1 or SW620 cells were first pre-treated (48h) with FE or HFE at 1/2 × IC_50_, 1 × IC_50_ and 2 × IC_50_. Non-treated cells were kept as controls. A total of 7500 DLD1 cells or 15,000 SW620 cells were plated in 96-well clear bottom black polystyrene plates. Then, cells were maintained for 6 h in complete media, without treatments, to allow for attachment. Then, an equal volume of the single-one-step reagent provided by the kit was added to each well and rocked for 15 min at room temperature. Cellular ATP content was measured by a luminescent plate reader.

### 4.6. Quantitative Real-Time Polymerase Chain Reaction (qRTPCR)

DLD1 and SW620 cells (0.35 × 10^6^ cells) were treated with FE or HFE for 48 hours at different doses: 1/2 IC_50_, 1× IC_50_, 2× IC_50_. Non-treated cells were kept as controls. For comparison and specificity of the targets, supercritical rosemary extract, whose molecular targets and mechanism of action has been previously characterised, was used (RE) [7,50]. 

Total RNA was extracted with Tri Reagent (Sigma). Then, 1 µg of RNA was reverse-transcribed with High Capacity RNA to cDNA Master Mix system (Life Technologies, Carlsbad, CA, USA). qPCR was performed in the 7900HT Real-Time PCR System (Life Technologies) using VeriQuest SYBR Green qPCR Master Mix (Affymetrix, Santa Clara, CA, USA). Appendix A shows the list and sequences of the oligos used and Taqman Probes used in this study. The 2^−ΔΔCt^ method was applied to calculate the relative gene expression [56].

### 4.7. Quantification of the Intracellular Neutral Lipid Content

Intracellular neutral lipid content was measured using Oil Red O staining as previously described by Ramírez-Zacarías et al. [57]. Briefly, the cells were washed gently twice with ice-cold PBS (pH 7.4) and fixed with 10% formalin at room temperature for at least 30 min to 1 h in room temperature. After removal of the 10% formalin, wells were washed with 60% isopropyl alcohol for 5 min and then washed exhaustively with PBS. Wells were allowed to dry completely before the addition of filtered Oil Red O solution for 30 min at room temperature. Stained oil droplets were extracted with 100% isopropanol for 10 min to quantify intracellular lipids, and absorbance was measured spectrophotometrically at 510 nm.

### 4.8. Statistical analysis

Results were analysed by ANOVA non-parametric with Bonferroni post hoc tests. Data were represented as mean ± S.E.M of at least three independent experiments. Statistical differences were defined as *p* < 0.05 (*); *p* < 0.01 (**); *p* < 0.005 (***); *p* < 0.001 (****). Statistical analysis was performed with Graph Pad Prism 8 statistical software (Version 8.0.0, GraphPad Software, San Diego, CA, USA).

## 5. Conclusions

As summary, extracts from fenugreek, but not from quinoa, show strong inhibitory effects on the viability of CRC cell lines. Additionally, the cytotoxicity of the saponin-rich extract from fenugreek is superior than its acid-hydrolyzed form, which is probably due to synergisms with other bioactive compounds contained in the extract. For the first time, it has been demonstrated that the cytotoxic effects of these extracts on CRC cells may be related to alterations in the energetic metabolism of the cells, specifically to a reduction in their mitochondrial oxidative phosphorylation and to the inhibition of their aerobic glycolysis. Lastly, saponin and sapogenin-rich extracts from fenugreek have proven to target in a different manner the molecular pathways related to lipid metabolism in both CRC cell lines studied. This outcome may have important implications on their potential against CRC in the context of a precision nutrition, as the saponin-rich extract from fenugreek would be useful for the upregulation of genes promoting systemic anabolic effects in patients developing cachexia, whereas the sapogenin-rich extract would be locally useful to inhibit lipid targets implicated in a poorer prognosis in CRC cells. In this sense, this study shows the relevance of considering the coexisting compounds of the extracts or their hydrolysis transformation as innovative strategies to augment the clinical therapeutic potential of the extracts, and the specific subgroup of patients where each extract would be more beneficial in the frame of precision nutrition.

## Figures and Tables

**Figure 1 cancers-12-03399-f001:**
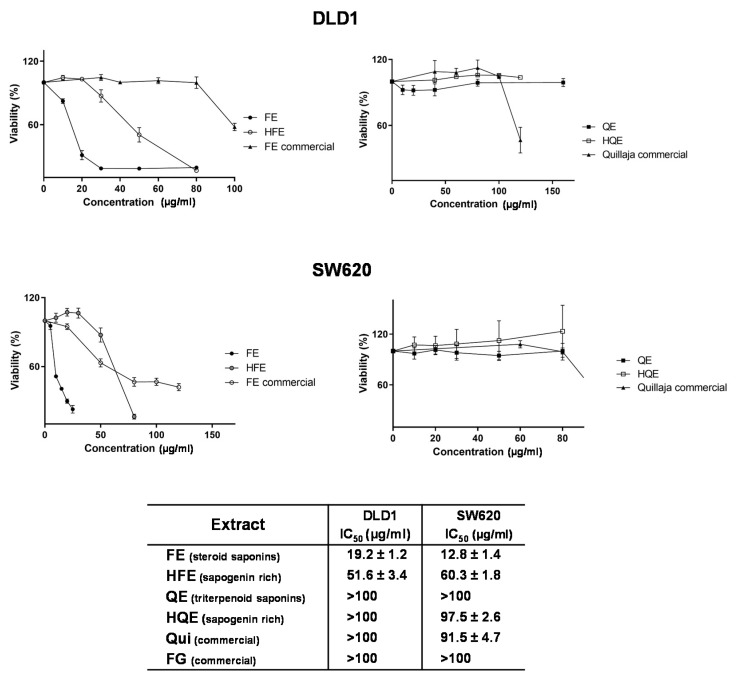
Dose–response curves and IC_50_ values of different saponin- and sapogenin- rich extracts on colorectal cancer (CRC) cell lines pretreated for 48 h with fenugreek and (FE), hydrolyzed fenugreek extract (HFE), quinoa (QE) and hydrolyzed quinoa extract (HQE) compared to control non-treated cells. For comparison purposes, two commercial extracts rich in steroid saponins (Fenfuro) and triterpenoid saponins (Quillaja) are shown.

**Figure 2 cancers-12-03399-f002:**
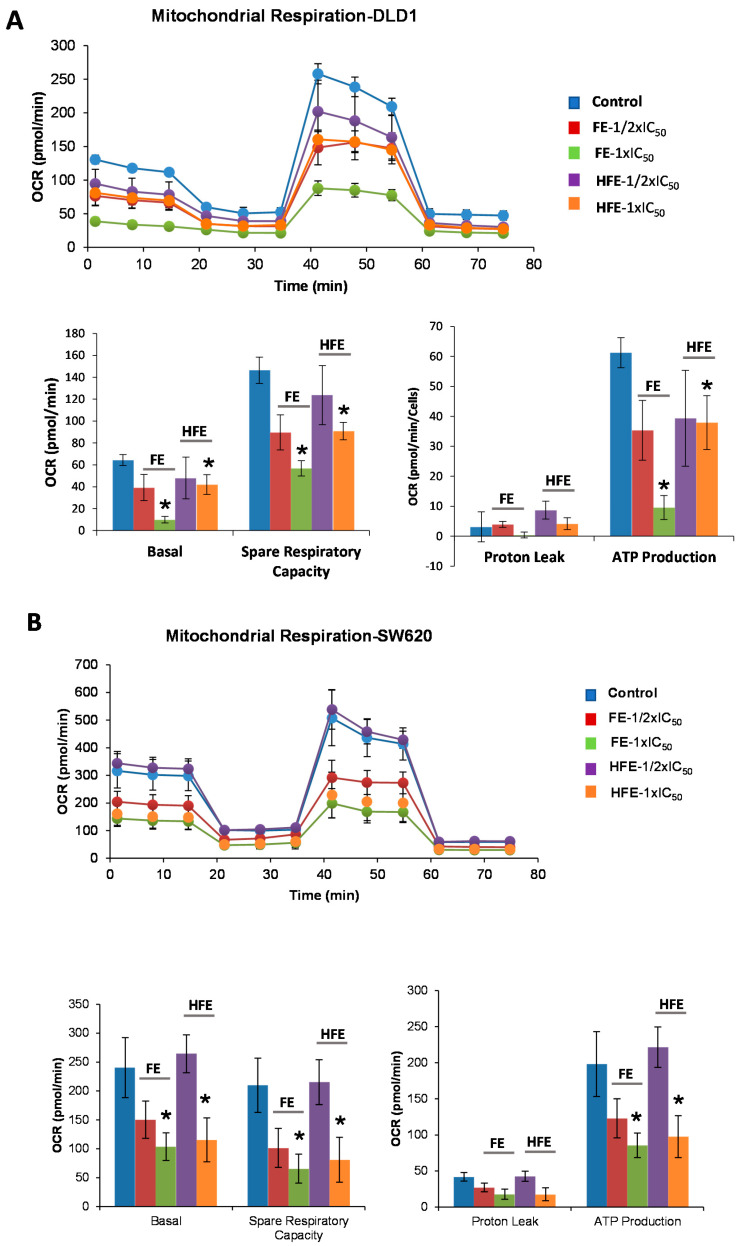
FE and HFE extracts inhibit mitochondrial oxidative phosphorylation of CRC cell lines. (**A**) MitoStress profile and quantification of basal oxygen consumption rates (OCR), Maximal Respiration Rate, ATP, Spare respiratory capacity, and Proton Leak of DLD1 CRC cells pre-treated with FE and HFE (at the indicated doses); (**B**) MitoStress profile and quantification of basal OCR, Maximal Respiration Rate, ATP production, Spare respiratory capacity, and Proton Leak of SW620 CRC cells pre-treated with FE and HFE (at the indicated doses). Asterisks indicate statistically significant differences (*p* < 0.05 (*)) relative to the control non-treated cells (6–8 replicates per condition).

**Figure 3 cancers-12-03399-f003:**
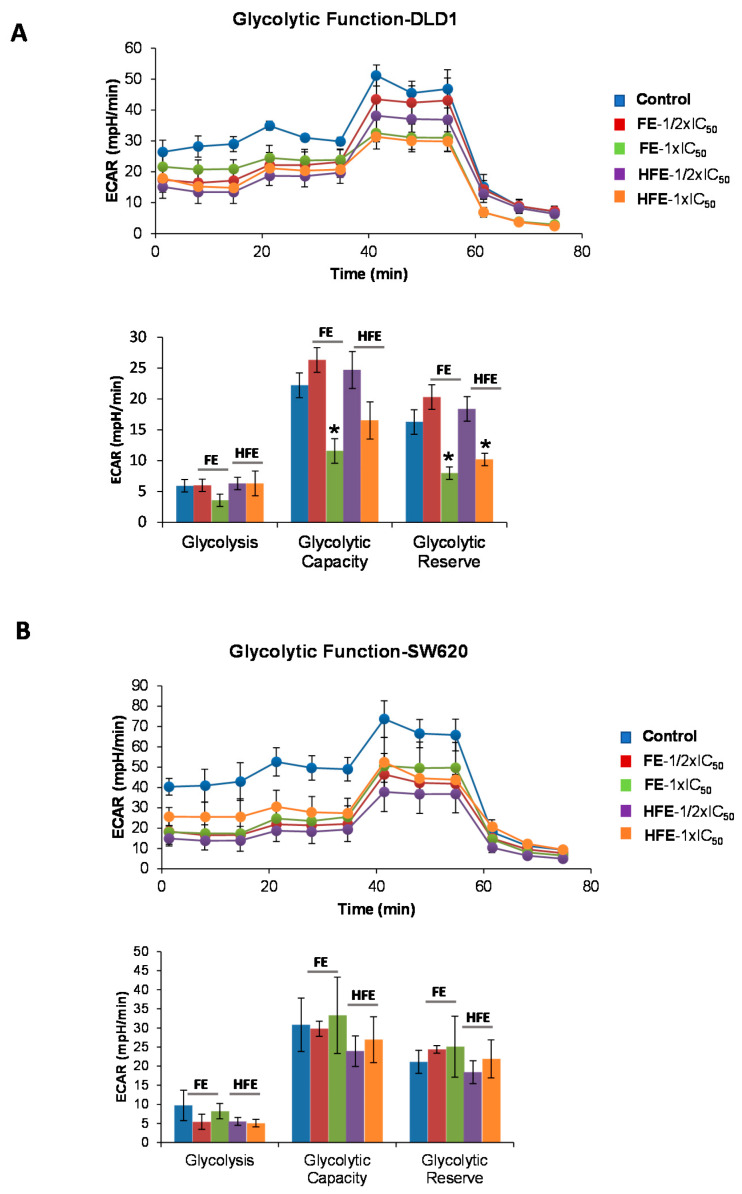
FE and HFE extracts inhibit aerobic glycolysis of CRC cell lines. (**A**) GlycoStress profile and quantification of basal ECAR, Maximal ECAR, and Glycolytic Reserve of DLD1 CRC cells pre-treated with FE and HFE (at the indicated doses). (**B**) GlycoStress profile and quantification of basal ECAR, Maximal ECAR, and Glycolytic Reserve of SW620 CRC cells pre-treated with FE and HFE (at the indicated doses). Asterisks indicate statistically significant differences (*p* < 0.05 (*)) relative to the control non-treated cells (6–8 replicates per condition).

**Figure 4 cancers-12-03399-f004:**
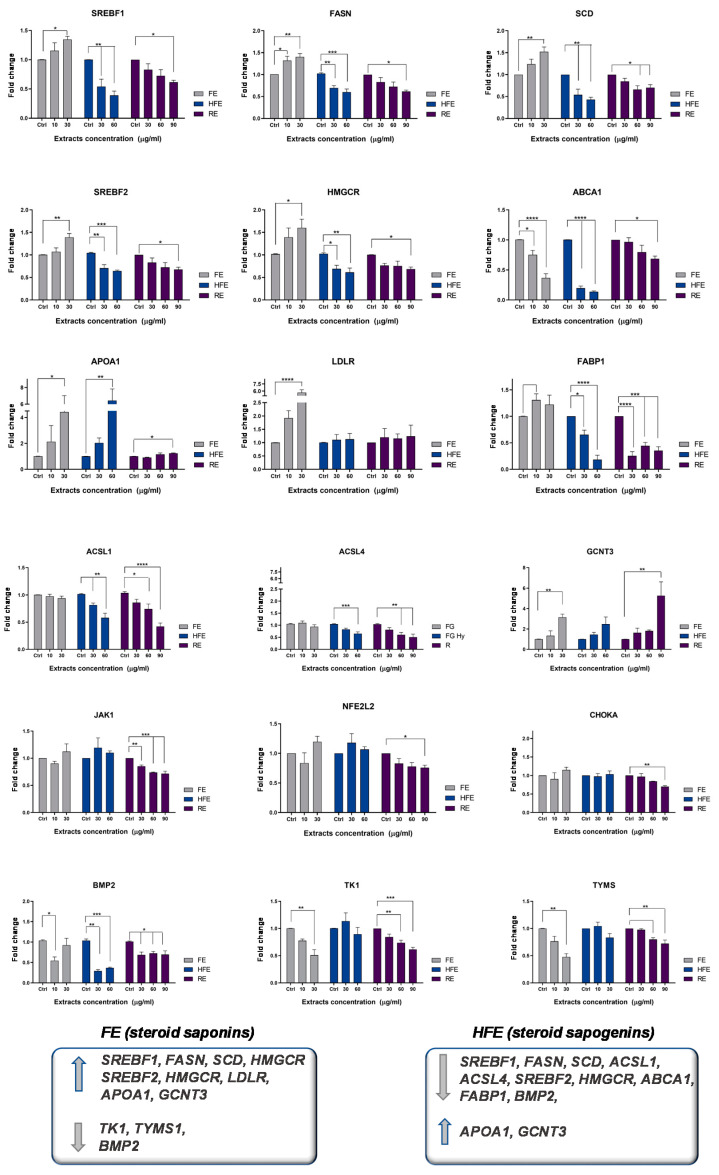
FE and HFE extracts differentially regulate metabolic targets in the DLD1 CRC cell line. For specificity of the molecular targets, the effects of a well-characterized supercritical rosemary extract (RE) is shown. Asterisks indicate statistically significant differences (*p* < 0.05 (*); *p* < 0.01 (**); *p* < 0.005 (***); *p* < 0.001 (****)) relative to the control non-treated cells (3–4 replicates, three independent experiments).

**Figure 5 cancers-12-03399-f005:**
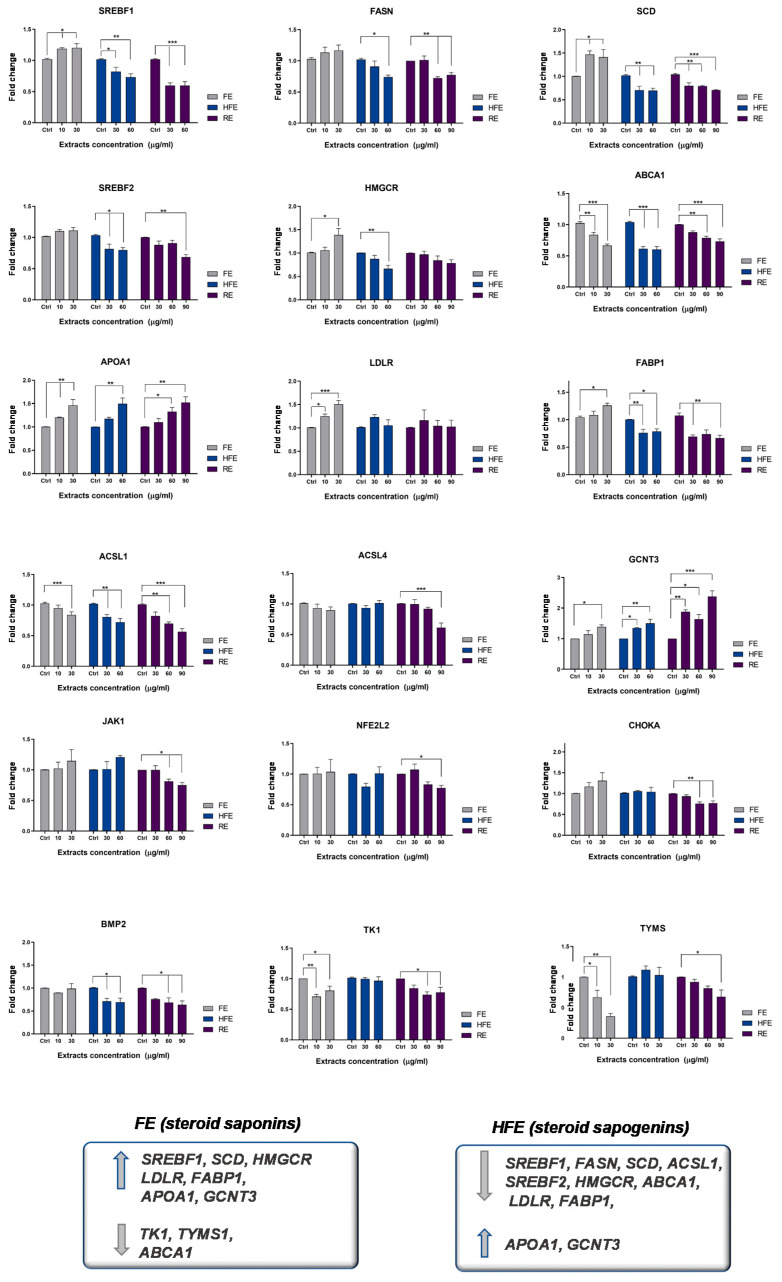
FE and HFE extracts differentially regulate metabolic targets in SW620 CRC cell line. For specificity of the molecular targets, the effects of a well-characterized supercritical rosemary extract (RE) is shown. Asterisks indicate statistically significant differences (*p* < 0.05 (*); *p* < 0.01 (**); *p* < 0.005 (***)) relative to the control non-treated cells (3–4 replicates, three independent experiments).

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
