# Peer review of "Saponin-Rich Extracts and Their Acid Hydrolysates Differentially Target Colorectal Cancer Metabolism in the Frame of Precision Nutrition"

_cancers, 2020, doi:10.3390/cancers12113399_

Round 1

Reviewer 1 Report

This study explored the effects of saponins-rich extracts from fenugreek and quinoa, named as FE and QE, respectively, and their hydrolyzed sapogenin-extract, named HFE and HQE, respectively on the colorectal cancer (CRC) cell lines. By means of cell bioenergetics analysis, authors found FE and HFE are effective neither QE or HQE is.

The fenugreek saponins and sapogenin-rich extracts target the molecular pathways related to lipid metabolism in the two CRC cell lines studied in different ways. It was clarified: FE inhibits the expression of TYMS1 and TK1, and synergizes with the chemotherapy drug 5-FluoroUracil-5FU, while HFE inhibits the target of lipid metabolism, resulting in a decrease in intracellular lipid content. In general, this study is interesting and provides novel results with significant information especially in the field of Precision Nutrition. However, there are some concerns should be clarified.

  1. It demonstrates that FE and HFE reduce mitochondrial oxidative phosphorylation and aerobic glycolysis at the cellular level. Does this result affect the performance of Hypoxia-inducible factor-1α (HIF-1α) and glucose transporter 1 (GLUT1)? As we know, the hypoxia-inducible factor 1α (HIF-1α) plays a major role in the tumor's response to hypoxia, and contributes to the tumor aggressiveness, invasiveness, and resistance to radiotherapy and chemotherapy.
  2. Hypoxia-inducible factor-1α (HIF-1α) and glucose transporter 1 (GLUT1) play an important role in the energy metabolism. What their roles in CRC should be researched and discussed.
  3. What’s your expect of applying FE / HFE to animal model?

Reviewer 2 Report

In this manuscript Gomez de Cedron et al. investigated the effects of saponin-rich extracts and their hydrolyzates from plants on colorectal cancer metabolism. They show that fenugreek extracts inhibit colorectal cancer cells viability, reduce both mitochondrial OXPHOS and glycolytic rates, and modulate lipid metabolism, which might have a potential to be used against colorectal cancer in the context of precision nutrition. The manuscript is clear and well-written, except for some minor English language mistakes.

Minor points:

  • The title of the manuscript is not specific and precise enough, I’d suggest it to be changed, in order to represent the data obtained clearly.
  • Line 19 – add “related” after (second) cancer.
  • Line 25 – delete “being”.
  • Lines 25 – 26 and throughout the manuscript – “mitochondrial oxidative respiration” should be changed to “mitochondrial oxidative phosphorylation” or “mitochondrial respiration”.
  • Line 69 – define “T2DM”, and in general, any abbreviation, when appearing for the first time.
  • Line 102 – “reviews” instead of “revisions”.
  • Line 123 – you cannot say “effects on inhibition”, delete “on inhibition”.
  • Line 140 – what do you mean by “former” saponins?
  • Line 143 – again, it is the “effect”, not the inhibition.
  • Figure 1 – you should correct the concentration unit in all the panels to mg/ml.
  • Line 147 – IC50 values are not of CRC lines, correct it.
  • Lines 167 – 168 – delete between the brackets, you don’t have to put methodology used in the subsection title.
  • Line 176 – delete “very”.
  • Line 204 – the same as for lines 167 – 8.
  • Line 208 – it is probably by mistake, correct “2 × IC50” into “1 × IC50”.
  • Lines 209 - 214 - this belongs to the Materials and methods section.
  • Line 218 – what is “d2”?
  • Lines 225 – 226 – the effects on SW620 cell line deserve more description.
  • Figure 4 – y axis title should be changed into “fold change”.
  • Lines 245 – 249 – seem to belong more to the Introduction than Results section.
  • Line 296 – add “the incidence of” after “The increase in”.
  • Line 300 – metabolic function, not dysfunction is altered.
  • Lines 428 – 429 – should be rephrased.
